# Understanding Instance-based Interpretability of Variational Auto-Encoders

**Zhifeng Kong**
Computer Science and Engineering
University of California San Diego
La Jolla, CA 92093
z4kong@eng.ucsd.edu

**Kamalika Chaudhuri**
Computer Science and Engineering
University of California San Diego
La Jolla, CA 92093
kamalika@cs.ucsd.edu

## Abstract

Instance-based interpretation methods have been widely studied for supervised learning methods as they help explain how black box neural networks predict. However, instance-based interpretations remain ill-understood in the context of unsupervised learning. In this paper, we investigate influence functions [Koh and Liang, 2017], a popular instance-based interpretation method, for a class of deep generative models called variational auto-encoders (VAE). We formally frame the counter-factual question answered by influence functions in this setting, and through theoretical analysis, examine what they reveal about the impact of training samples on classical unsupervised learning methods. We then introduce VAE-TracIn, a computationally efficient and theoretically sound solution based on Pruthi et al. [2020], for VAEs. Finally, we evaluate VAE-TracIn on several real world datasets with extensive quantitative and qualitative analysis.

## 1 Introduction

Instance-based interpretation methods have been popular for supervised learning as they help explain why a model makes a certain prediction and hence have many applications [Barshan et al., 2020, Basu et al., 2020, Chen et al., 2020, Ghorbani and Zou, 2019, Hara et al., 2019, Harutyunyan et al., 2021, Koh and Liang, 2017, Koh et al., 2019, Pruthi et al., 2020, Yeh et al., 2018, Yoon et al., 2020]. For a classifier and a test sample $z$, an instance-based interpretation ranks all training samples $x$ according to an interpretability score between $x$ and $z$. Samples with high (low) scores are considered positively (negatively) important for the prediction of $z$.

However, in the literature of unsupervised learning especially generative models, instance-based interpretations are much less understood. In this work, we investigate instance-based interpretation methods for unsupervised learning based on influence functions [Cook and Weisberg, 1980, Koh and Liang, 2017]. In particular, we theoretically analyze certain classical non-parametric and parametric methods. Then, we look at a canonical deep generative model, variational auto-encoders (VAE, [Higgins et al., 2016, Kingma and Welling, 2013]), and explore some of the applications.

The first challenge is framing the counter-factual question for unsupervised learning. For instance-based interpretability in supervised learning, the counter-factual question is "which training samples are most responsible for the prediction of a test sample?" – which heavily relies on the label information. However, there is no label in unsupervised learning. In this work, we frame the counter-factual question for unsupervised learning as "which training samples are most responsible for increasing the likelihood (or reducing the loss when likelihood is not available) of a test sample?" We show that influence functions can answer this counter-factual question. Then, we examine influence functions for several classical unsupervised learning methods. We present theory and intuitions on how influence functions relate to likelihood and pairwise distances.

35th Conference on Neural Information Processing Systems (NeurIPS 2021).

The second challenge is how to compute influence functions in VAE. The first difficulty here is that the VAE loss of a test sample involves an expectation over the encoder, so the actual influence function cannot be precisely computed. To deal with this problem, we use the empirical average of influence functions, and prove a concentration bound of the empirical average under mild conditions. Another difficulty is computation. The influence function involves inverting the Hessian of the loss with respect to all parameters, which involves massive computation for big neural networks with millions of parameters. To deal with this problem, we adapt a first-order estimate of the influence function called TracIn [Pruthi et al., 2020] to VAE. We call our method VAE-TracIn. It is fast because $(i)$ it does not involve the Hessian, and $(ii)$ it can accelerate computation with only a few checkpoints.

We begin with a sanity check that examines whether training samples have the highest influences over themselves, and show VAE-TracIn passes it. We then evaluate VAE-TracIn on several real world datasets. We find high (low) self influence training samples have large (small) losses. Intuitively, high self influence samples are hard to recognize or visually high-contrast, while low self influence ones share similar shapes or background. These findings lead to an application on unsupervised data cleaning, as high self influence samples are likely to be outside the data manifold. We then provide quantitative and visual analysis on influences over test data. We call high and low influence samples *proponents* and *opponents*, respectively. [1] We find in certain cases both proponents and opponents are similar samples from the same class, while in other cases proponents have large norms.

We consider VAE-TracIn as a general-purpose tool that can potentially help understand many aspects in the unsupervised setting, including $(i)$ detecting underlying memorization, bias or bugs [Feldman and Zhang, 2020] in unsupervised learning, $(ii)$ performing data deletion [Asokan and Seelamantula, 2020, Izzo et al., 2021] in generative models, and $(iii)$ examining training data without label information.

We make the following contributions in this paper.

- We formally frame instance-based interpretations for unsupervised learning.
- We examine influence functions for several classical unsupervised learning methods.
- We present VAE-TracIn, an instance-based interpretation method for VAE. We provide both theoretical and empirical justification to VAE-TracIn.
- We evaluate VAE-TracIn on several real world datasets. We provide extensive quantitative analysis and visualization, as well as an application on unsupervised data cleaning.

## 1.1 Related Work

There are two lines of research on instance-based interpretation methods for supervised learning.

The first line of research frames the following counter-factual question: which training samples are most responsible for the prediction of a particular test sample $z$? This is answered by designing an interpretability score that measures the importance of training samples over $z$ and selecting those with the highest scores. Many scores and their approximations have been proposed [Barshan et al., 2020, Basu et al., 2020, Chen et al., 2020, Hara et al., 2019, Koh and Liang, 2017, Koh et al., 2019, Pruthi et al., 2020, Yeh et al., 2018]. Specifically, Koh and Liang [2017] introduce the influence function (IF) based on the terminology in robust statistics [Cook and Weisberg, 1980]. The intuition is removing an important training sample of $z$ should result in a huge increase of its test loss. Because the IF is hard to compute, Pruthi et al. [2020] propose TracIn, a fast first-order approximation to IF.

Our paper extends the counter-factual question to unsupervised learning where there is no label. We ask: which training samples are most responsible for increasing the likelihood (or reducing the loss) of a test sample? In this paper, we propose VAE-TracIn, an instance-based interpretation method for VAE [Higgins et al., 2016, Kingma and Welling, 2013] based on TracIn and IF.

The second line of research considers a different counter-factual question: which training samples are most responsible for the overall performance of the model (e.g. accuracy)? This is answered by designing an interpretability score for each training sample. Again many scores have been proposed [Ghorbani and Zou, 2019, Harutyunyan et al., 2021, Yoon et al., 2020]. Terashita et al. [2021] extend this framework to a specific unsupervised model called generative adversarial networks [Goodfellow

---

[1]There are different names in the literature, such as helpful/harmful samples [Koh and Liang, 2017], excitatory/inhibitory points [Yeh et al., 2018], and proponents/opponents [Pruthi et al., 2020].

et al., 2014]. They measure influences of samples on several evaluation metrics, and discard samples that harm these metrics. Our paper is orthogonal to these works.

The instance-based interpretation methods lead to many applications in various areas including adversarial learning, data cleaning, prototype selection, data summarization, and outlier detection [Barshan et al., 2020, Basu et al., 2020, Chen et al., 2020, Feldman and Zhang, 2020, Ghorbani and Zou, 2019, Hara et al., 2019, Harutyunyan et al., 2021, Khanna et al., 2019, Koh and Liang, 2017, Pruthi et al., 2020, Suzuki et al., 2021, Ting and Brochu, 2018, Ye et al., 2021, Yeh et al., 2018, Yoon et al., 2020]. In this paper, we apply the proposed VAE-TracIn to an unsupervised data cleaning task.

Prior works on interpreting generative models analyze their latent space via measuring disentanglement, explaining and visualizing representations, or analysis in an interactive interface [Alvarez-Melis and Jaakkola, 2018, Bengio et al., 2013, Chen et al., 2016, Desjardins et al., 2012, Kim and Mnih, 2018, Olah et al., 2017, 2018, Ross et al., 2021]. These latent space analysis are complementary to the instance-based interpretation methods in this paper.

## 2 Instance-based Interpretations

Let $X = \{x_i\}_{i=1}^N \in \mathbb{R}^d$ be the training set. Let $\mathcal{A}$ be an algorithm that takes $X$ as input and outputs a model that describes the distribution of $X$. $\mathcal{A}$ can be a density estimator or a generative model. Let $L(X; \mathcal{A}) = \frac{1}{N} \sum_{i=1}^N \ell(x_i; \mathcal{A}(X))$ be a loss function. Then, the influence function of a training data $x_i$ over a test data $z \in \mathbb{R}^d$ is the loss of $z$ computed from the model trained without $x_i$ minus that computed from the model trained with $x_i$. If the difference is large, then $x_i$ should be very influential for $z$. Formally, the influence function is defined below.

**Definition 1** (Influence functions [Koh and Liang, 2017]). *Let $X_{-i} = X \setminus \{x_i\}$. Then, the influence of $x_i$ over $z$ is defined as $\mathrm{IF}_{X,\mathcal{A}}(x_i, z) = \ell(z; \mathcal{A}(X_{-i})) - \ell(z; \mathcal{A}(X))$. If $\mathrm{IF}_{X,\mathcal{A}}(x_i, z) > 0$, we say $x_i$ is a proponent of $z$; otherwise, we say $x_i$ is an opponent of $z$.*

For big models $\mathcal{A}$ such as deep neural networks, doing retraining and obtaining $\mathcal{A}(X_{-i})$ can be expensive. The following TracIn score is a fast approximation to IF.

**Definition 2** (TracIn scores [Pruthi et al., 2020]). *Suppose $\mathcal{A}(X)$ is obtained by minimizing $L(X; \mathcal{A})$ via stochastic gradient descent. Let $\{\theta_{[c]}\}_{c=1}^C$ be $C$ checkpoints during the training procedure. Then, the estimated influence of $x_i$ over $z$ is defined as $\mathrm{TracIn}_{X,\mathcal{A}}(x_i, z) = \sum_{c=1}^C \nabla\ell(x_i; \theta_{[c]})^\top \nabla\ell(z; \theta_{[c]})$.*

We are also interested in the influence of a training sample over itself. Formally, we define this quantity as the self influence of $x$, or Self-$\mathrm{IF}_{X,\mathcal{A}}(x) = \mathrm{IF}_{X,\mathcal{A}}(x, x)$. In supervised learning, self influences provide rich information about memorization properties of training samples. Intuitively, high self influence samples are atypical, ambiguous or mislabeled, while low self influence samples are typical [Feldman and Zhang, 2020].

## 3 Influence Functions for Classical Unsupervised Learning

In this section, we analyze influence functions for unsupervised learning. The goal is to provide intuition on what influence functions should tell us in the unsupervised setting. Specifically, we look at three classical methods: the non-parametric $k$-nearest-neighbor ($k$-NN) density estimator, the non-parametric kernel density estimator (KDE), and the parametric Gaussian mixture models (GMM). We let the loss function $\ell$ to be the negative log-likelihood: $\ell(z) = -\log p(z)$.

**The $k$-Nearest-Neighbor ($k$-NN) density estimator.** The $k$-NN density estimator is defined as $p_{k\mathrm{NN}}(x; X) = k/(N V_d R_k(x; X)^d)$, where $R_k(x; X)$ is the distance between $x$ and its $k$-th nearest neighbor in $X$ and $V_d$ is the volume of the unit ball in $\mathbb{R}^d$. Then, we have the following influence function for the $k$-NN density estimator:

$$\mathrm{IF}_{X,k\mathrm{NN}}(x_i, z) = \log\frac{N-1}{N} + \begin{cases} d\log\frac{R_{k+1}(z; X)}{R_k(z; X)} & \|x_i - z\| \leq R_k(z; X) \\ 0 & \text{otherwise} \end{cases}. \quad (1)$$

See Appendix A.1.1 for proof. Note, when $z$ is fixed, there are only two possible values for training data influences: $\log\frac{N-1}{N}$ and $\log\frac{N-1}{N} + d\log\frac{R_{k+1}(z; X)}{R_k(z; X)}$. As for Self-$\mathrm{IF}_{X,k\mathrm{NN}}(x_i)$, samples with

the largest self influences are those with the largest $\frac{R_{k+1}(x_i;X)}{R_k(x_i;X)}$. Intuitively, these samples belong to a cluster of size exactly $k$, and the cluster is far away from other samples.

**Kernel Density Estimators (KDE).** The KDE is defined as $p_{\text{KDE}}(x;X) = \frac{1}{N}\sum_{i=1}^{N} K_\sigma(x - x_i)$, where $K_\sigma$ is the Gaussian $\mathcal{N}(0, \sigma^2 I)$. Then, we have the following influence function for KDE:

$$\text{IF}_{X,\text{KDE}}(x_i, z) = \log\frac{N-1}{N} + \log\left(1 + \frac{\frac{1}{N}K_\sigma(z - x_i)}{p_{\text{KDE}}(z;X) - \frac{1}{N}K_\sigma(z - x_i)}\right). \tag{2}$$

See Appendix A.1.2 for proof. For a fixed $z$, an $x_i$ with larger $\|z - x_i\|$ has a higher influence over $z$. Therefore, the strongest proponents of $z$ are those closest to $z$ in the $\ell_2$ distance, and the strongest opponents are the farthest. As for Self-$\text{IF}_{X,\text{KDE}}(x_i)$, samples with the largest self influences are those with the least likelihood $p_{\text{KDE}}(x_i; X)$. Intuitively, these samples locate at very sparse regions and have few nearby samples. On the other hand, samples with the largest likelihood $p_{\text{KDE}}(x_i; X)$, or those in the high density area, have the least self influences.

**Gaussian Mixture Models (GMM).** As there is no closed-form expression for general GMM, we make the following well-separation assumption to simplify the problem.

**Assumption 1.** $X = \bigcup_{k=0}^{K-1} X_k$ where each $X_k$ is a cluster. We assume these clusters are well-separated: $\min\{\|x - x'\| : x \in X_k, x' \in X_{k'}\} \gg \max\{\|x - x'\| : x, x' \in X_k\}$.

Let $|X_k| = N_k$ and $N = \sum_{k=0}^{K-1} N_k$. For $x \in \mathbb{R}^d$, let $k = \arg\min_i d(x, X_i)$. Then, we define the well-separated spherical GMM (WS-GMM) of $K$ mixtures as $p_{\text{WS-GMM}}(x) = \frac{N_k}{N}\mathcal{N}(x; \mu_k, \sigma_k^2 I)$, where the parameters are given by the maximum likelihood estimates

$$\mu_k = \frac{1}{N_k}\sum_{x \in X_k} x, \ \sigma_k^2 = \frac{1}{N_k d}\sum_{x \in X_k} \|x - \mu_k\|^2 = \frac{1}{N_k d}\sum_{x \in X_k} x^\top x - \frac{1}{d}\mu_k^\top \mu_k. \tag{3}$$

For conciseness, we let test sample $z$ from cluster zero: $z \in \text{conv}(X_0)$. Then, we have the following influence function for WS-GMM. If $x_i \notin X_0$, $\text{IF}_{X,\text{WS-GMM}}(x_i, z) = -\frac{1}{N} + \mathcal{O}(N^{-2})$. Otherwise,

$$\text{IF}_{X,\text{WS-GMM}}(x_i, z) = \frac{d+2}{2N_0} + \frac{1}{2N_0\sigma_0^2}\left(\frac{\|z - \mu_0\|^2}{\sigma_0^2} - \|z - x_i\|^2\right) - \frac{1}{N} + \mathcal{O}(N_0^{-2}). \tag{4}$$

See Appendix A.1.3 for proof. A surprising finding is that some $x_i \in X_0$ may have very negative influences over $z$ (i.e. strong opponents of $z$ are from the same class). This happens with high probability if $\|z - x_i\|^2 \gtrsim (1 + \sigma_0^2)d + 2\sigma_0^2$ for large dimension $d$. Next, we compute the self influence of an $x_i \in X_k$. According to (4),

$$\text{Self-IF}_{X,\text{WS-GMM}}(x_i) = \frac{d+2}{2N_k} + \frac{\|x_i - \mu_k\|^2}{2N_k\sigma_k^4} - \frac{1}{N} + \mathcal{O}(N_k^{-2}). \tag{5}$$

Within each cluster $X_k$, samples far away to the cluster center $\mu_k$ have large self influences and vice versa. Across the entire dataset, samples in cluster $X_k$ whose $N_k$ or $\sigma_k$ is small tend to have large self influences, which is very different from $k$-NN or KDE.

### 3.1 Summary

We summarize the intuitions of influence functions in classical unsupervised learning in Table 4. Among these methods, the strong proponents are all nearest samples, but self influences and strong opponents are quite different. We then visualize an example of six clusters of 2D points in Figure 5 in Appendix B.1. In Figure 6, We plot the self influences of these data points under different density estimators. For a test data point $z$, we plot influences of all data points over $z$ in Figure 7.

## 4 Instance-based Interpretations for Variational Auto-encoders

In this section, we show how to compute influence functions for a class of deep generative models called variational auto-encoders (VAE). Specifically, we look at $\beta$-VAE [Higgins et al., 2016] defined below, which generalizes the original VAE by Kingma and Welling [2013].

**Definition 3** ($\beta$-VAE [Higgins et al., 2016]). *Let $d_{\text{latent}}$ be the latent dimension. Let $P_\phi : \mathbb{R}^{d_{\text{latent}}} \to \mathbb{R}^+$ be the decoder and $Q_\psi : \mathbb{R}^d \to \mathbb{R}^+$ be the encoder, where $\phi$ and $\psi$ are the parameters of the networks. Let $\theta = [\phi, \psi]$. Let the latent distribution $P_{\text{latent}}$ be $\mathcal{N}(0, I)$. For $\beta > 0$, the $\beta$-VAE model minimizes the following loss:*

$$L_\beta(X; \theta) = \mathbb{E}_{x \sim X} \ell_\beta(x; \theta) = \beta \cdot \mathbb{E}_{x \sim X} \text{KL} \left( Q_\psi(\cdot|x) \| P_{\text{latent}} \right) - \mathbb{E}_{x \sim X} \mathbb{E}_{\xi \sim Q_\psi(\cdot|x)} \log P_\phi(x|\xi). \quad (6)$$

In practice, the encoder $Q = Q_\psi$ outputs two vectors, $\mu_Q$ and $\sigma_Q$, so that $Q(\cdot|x) = \mathcal{N}(\mu_Q(x), \text{diag}(\sigma_Q(x))^2 I)$. The decoder $P = P_\phi$ outputs a vector $\mu_P$ so that $\log P(x|\xi)$ is a constant times $\|\mu_P(\xi) - x\|^2$ plus a constant.

Let $\mathcal{A}$ be the $\beta$-VAE that returns $\mathcal{A}(X) = \arg\min_\theta L_\beta(X; \theta)$. Let $\theta^* = \mathcal{A}(X)$ and $\theta^*_{-i} = \mathcal{A}(X_{-i})$. Then, the influence function of $x_i$ over a test point $z$ is $\ell_\beta(z; \theta^*_{-i}) - \ell_\beta(z; \theta^*)$, which equals to

$$
\begin{aligned}
\text{IF}_{X,\text{VAE}}(x_i, z) &= \beta \left( \text{KL}\left( Q_{\psi^*_{-i}}(\cdot|z) \| P_{\text{latent}} \right) - \text{KL}\left( Q_{\psi^*}(\cdot|z) \| P_{\text{latent}} \right) \right) \\
&\quad - \left( \mathbb{E}_{\xi \sim Q_{\psi^*_{-i}}(\cdot|z)} \log P_{\phi^*_{-i}}(z|\xi) - \mathbb{E}_{\xi \sim Q_{\psi^*}(\cdot|z)} \log P_{\phi^*}(z|\xi) \right).
\end{aligned}
\quad (7)
$$

**Challenge.** The first challenge is that IF in (7) involves an expectation over the encoder, so it cannot be precisely computed. To solve the problem, we compute the empirical average of the influence function over $m$ samples. In **Theorem** 1, we theoretically prove that the empirical influence function is close to the actual influence function with high probability when $m$ is properly selected. The second challenge is that IF is hard to compute. To solve this problem, in Section 4.1, we propose VAE-TracIn, a computationally efficient solution to VAE.

**A probabilistic bound on influence estimates.** Let $\hat{\text{IF}}^{(m)}_{X,\text{VAE}}$ be the empirical average of the influence function over $m$ i.i.d. samples. We have the following result.

**Theorem 1** (Error bounds on influence estimates (informal, see formal statement in **Theorem** 4)). *Under mild conditions, for any small $\epsilon > 0$ and $\delta > 0$, there exists an $m = \Theta\left(\frac{1}{\epsilon^2 \delta}\right)$ such that*

$$\text{Prob}\left( \left| \text{IF}_{X,\text{VAE}}(x_i, z) - \hat{\text{IF}}^{(m)}_{X,\text{VAE}}(x_i, z) \right| \geq \epsilon \right) \leq \delta. \quad (8)$$

Formal statements and proofs are in Appendix A.2.

## 4.1 VAE-TracIn

In this section, we introduce VAE-TracIn, a computationally efficient interpretation method for VAE. VAE-TracIn is built based on TracIn (**Definition** 2). According to (6), the gradient of the loss $\ell_\beta(x; \theta)$ can be written as $\nabla_\theta \ell_\beta(x; \theta) = [\nabla_\phi \ell_\beta(x; \theta)^\top, \nabla_\psi \ell_\beta(x; \theta)^\top]^\top$, where

$$
\begin{aligned}
\nabla_\phi \ell_\beta(x; \theta) &= \mathbb{E}_{\xi \sim Q_\psi(\cdot|x)} \left( -\nabla_\phi \log P_\phi(x|\xi) \right) =: \mathbb{E}_{\xi \sim Q_\psi(\cdot|x)} U(x, \xi, \phi, \psi), \text{ and} \\
\nabla_\psi \ell_\beta(x; \theta) &= \mathbb{E}_{\xi \sim Q_\psi(\cdot|x)} \nabla_\psi \log Q_\psi(\xi|x) \left( \beta \log \frac{Q_\psi(\xi|x)}{P_{\text{latent}}(\xi)} - \log P_\phi(x|\xi) \right) \\
&=: \mathbb{E}_{\xi \sim Q_\psi(\cdot|x)} V(x, \xi, \phi, \psi).
\end{aligned}
\quad (9)
$$

The derivations are based on the Stochastic Gradient Variational Bayes estimator [Kingma and Welling, 2013], which offers low variance [Rezende et al., 2014]. See Appendix A.3 for full details of the derivation. We estimate the expectation $\mathbb{E}_\xi$ by averaging over $m$ i.i.d. samples. Then, for a training data $x$ and test data $z$, the VAE-TracIn score of $x$ over $z$ is computed as

$$
\begin{aligned}
\text{VAE-TracIn}(x, z) &= \sum_{c=1}^{C} \left( \frac{1}{m} \sum_{j=1}^{m} U(x, \xi_{j,[c]}, \phi_{[c]}, \psi_{[c]}) \right)^\top \left( \frac{1}{m} \sum_{j=1}^{m} U(z, \xi'_{j,[c]}, \phi_{[c]}, \psi_{[c]}) \right) \\
&\quad + \sum_{c=1}^{C} \left( \frac{1}{m} \sum_{j=1}^{m} V(x, \xi_{j,[c]}, \phi_{[c]}, \psi_{[c]}) \right)^\top \left( \frac{1}{m} \sum_{j=1}^{m} V(z, \xi'_{j,[c]}, \phi_{[c]}, \psi_{[c]}) \right),
\end{aligned}
\quad (10)
$$

where the notations $U, V$ are from (9), $\theta_{[c]} = [\phi_{[c]}, \psi_{[c]}]$ is the $c$-th checkpoint, $\{\xi_{j,[c]}\}_{j=1}^m$ are i.i.d. samples from $Q_{\psi_{[c]}}(\cdot|x)$, and $\{\xi'_{j,[c]}\}_{j=1}^m$ are i.i.d. samples from $Q_{\psi_{[c]}}(\cdot|z)$.

Table 1: Sanity check on the frequency that a training sample is the most influential one over itself. Results on MNIST, CIFAR, and the averaged result on CIFAR subclasses are reported.

| MNIST | | CIFAR | | Averaged CIFAR subclass |
| --- | --- | --- | --- | --- |
| $d_{\text{latent}} = 64$ | $d_{\text{latent}} = 128$ | $d_{\text{latent}} = 64$ | $d_{\text{latent}} = 128$ | $d_{\text{latent}} = 64$ |
| 0.992 | 1.000 | 0.609 | 0.602 | 0.998 |

**Connections between VAE-TracIn and influence functions [Koh and Liang, 2017].** Koh and Liang [2017] use the second-order (Hessian-based) approximation to the change of loss under the assumption that the loss function is convex. The TracIn algorithm [Pruthi et al., 2020] uses the first-order (gradient-based) approximation to the change of loss during the training process under the assumption that (stochastic) gradient descent is the optimizer. We expect these methods to give similar results in the ideal situation. However, we implemented the method by Koh and Liang [2017] and found it to be inaccurate for VAE. A possible reason is that the Hessian vector product used to approximate the second order term is unstable.

**Complexity of VAE-TracIn.** The run-time complexity of VAE-TracIn is linear in the number of samples ($N$), checkpoints ($C$), and network parameters ($|\theta|$).

## 5 Experiments

In this section, we aim to answer the following questions.

- Does VAE-TracIn pass a sanity check for instance-based interpretations?
- Which training samples have the highest and lowest self influences, respectively?
- Which training samples have the highest influences over (i.e. are strong proponents of) a test sample? Which have the lowest influences over it (i.e. are its strong opponents)?

These questions are examined in experiments on the MNIST [LeCun et al., 2010] and CIFAR-10 [Krizhevsky et al., 2009] datasets.

### 5.1 Sanity Check

**Question.** Does VAE-TracIn find the most influential training samples? In a perfect instance-based interpretation for a good model, training samples should have large influences over themselves. As a sanity check, we examine if training samples are the strongest proponents over themselves. This is analogous to the identical subclass test by Hanawa et al. [2020].

**Methodology.** We train separate VAE models on MNIST, CIFAR, and each CIFAR subclass (the set of five thousand CIFAR samples in each class). For each model, we examine the frequency that a training sample is the most influential one among all samples over itself. Due to computational limits we examine the first 128 samples. The results for MNIST, CIFAR, and the averaged result for CIFAR subclasses are reported in Table 1. Detailed results for CIFAR subclasses are in Appendix B.3.

**Results.** The results indicate that VAE-TracIn can find the most influential training samples in MNIST and CIFAR subclasses. This is achieved even under the challenge that many training samples are very similar to each other. The results for CIFAR is possibly due to underfitting as it is challenging to train a good VAE on this dataset. Note, the same VAE architecture is trained on CIFAR subclasses.

**Visualization.** We visualize some correctly and incorrectly identified strongest proponents in Figure 1. On MNIST or CIFAR subclasses, even if a training sample is not exactly the strongest proponent of itself, it still ranks very high in the order of influences.

### 5.2 Self Influences

**Question.** Which training samples have the highest and lowest self influences, respectively? Self influences provide rich information about properties of training samples such as memorization. In supervised learning, high self influence samples can be atypical, ambiguous or mislabeled, while

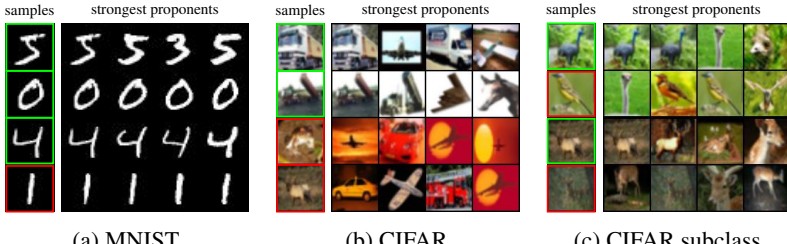

| (a) MNIST | (b) CIFAR | (c) CIFAR subclass |

Figure 1: Certain training samples and their strongest proponents in the training set (sorted from left to right). A sample $x_i$ is marked in green box if it is more influential than other samples over itself (i.e. it is the strongest proponent of itself) and otherwise in red box.

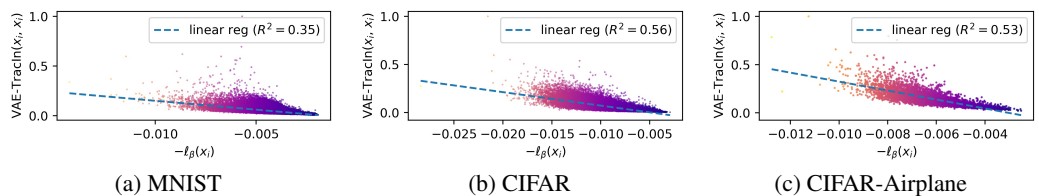

| (a) MNIST | (b) CIFAR | (c) CIFAR-Airplane |

Figure 2: Scatter plots of self influences versus negative losses of all training samples in several datasets. The linear regressors show that high self influence samples have large losses.

low self influence samples are typical [Feldman and Zhang, 2020]. We examine what self influences reveal in VAE.

**Methodology.** We train separate VAE models on MNIST, CIFAR, and each CIFAR subclass. We then compute the self influences and losses of each training sample. We show the scatter plots of self influences versus negative losses in Figure 2. [2] We fit linear regression models to these points and report the $R^2$ scores of the regressors. More comparisons including the marginal distributions and the joint distributions can be found in Appendix B.4 and Appendix B.5.

**Results.** We find the self influence of a training sample $x_i$ tends to be large when its loss $\ell_\beta(x_i)$ is large. This finding in VAE is consistent with KDE and GMM (see Figure 6). In supervised learning, Pruthi et al. [2020] find high self influence samples come from densely populated areas while low self influence samples come from sparsely populated areas. Our findings indicate significant difference between supervised and unsupervised learning in terms of self influences under certain scenarios.

**Visualization.** We visualize high and low self influence samples in Figure 3 (more visualizations in Appendix B.5). High self influence samples are either hard to recognize or visually high-contrast, while low self influence samples share similar shapes or background. These visualizations are consistent with the memorization analysis by Feldman and Zhang [2020] in the supervised setting. We also notice that there is a concurrent work connecting self influences on log-likelihood and memorization properties in VAE through cross validation and retraining [van den Burg and Williams, 2021]. Our quantitative and qualitative results are consistent with their results.

**Application on unsupervised data cleaning.** A potential application on unsupervised data cleaning is to use self influences to detect unlikely samples and let a human expert decide whether to discard them before training. The unlikely samples may include noisy samples, contaminated samples, or incorrectly collected samples due to bugs in the data collection process. For example, they could be unrecognizable handwritten digits in MNIST or objects in CIFAR. Similar approaches in supervised learning use self influences to detect mislabeled data [Koh and Liang, 2017, Pruthi et al., 2020, Yeh et al., 2018] or memorized samples [Feldman and Zhang, 2020]. We extend the application of self influences to scenarios where there are no labels.

---

[2]We use the negative loss because it relates to the log-likelihood of $x_i$: when $\beta = 1$, $-\ell_\beta(x) \leq \log p(x)$.

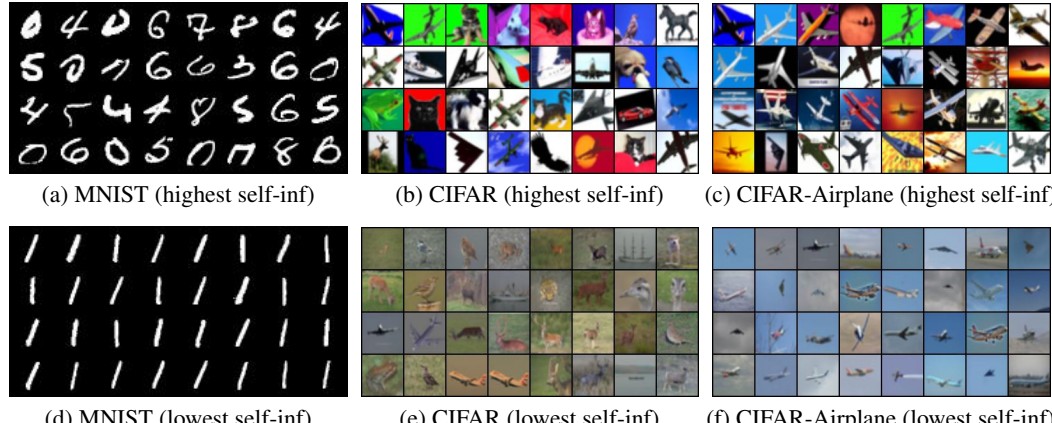

| (a) MNIST (highest self-inf) | (b) CIFAR (highest self-inf) | (c) CIFAR-Airplane (highest self-inf) |
| (d) MNIST (lowest self-inf) | (e) CIFAR (lowest self-inf) | (f) CIFAR-Airplane (lowest self-inf) |

Figure 3: High and low self influence samples from several datasets. High self influence samples are hard to recognize or high-contrast. Low self influence samples share similar shapes or background.

To test this application, we design an experiment to see if self influences can find a small amount of extra samples added to the original dataset. The extra samples are from other datasets: 1000 EMNIST [Cohen et al., 2017] samples for MNIST, and 1000 CelebA [Liu et al., 2015] samples for CIFAR, respectively. In Figure 15, we plot the detection curves to show fraction of extra samples found when all samples are sorted in the self influence order. The area under these detection curves (AUC) are 0.887 in the MNIST experiment and 0.760 in the CIFAR experiment. [3] Full results and more comparisons can be found in Appendix B.6. The results indicate that extra samples generally have higher self influences than original samples, so it has much potential to apply VAE-TracIn to unsupervised data cleaning.

### 5.3 Influences over Test Data

**Question.** Which training samples are strong proponents or opponents of a test sample, respectively? Influences over a test sample $z$ provide rich information about the relationship between training data and $z$. In supervised learning, strong proponents help the model correctly predict the label of $z$ while strong opponents harm it. Empirically, strong proponents are visually similar samples from the same class, while strong opponents tend to confuse the model [Pruthi et al., 2020]. In unsupervised learning, we expect that strong proponents increase the likelihood of $z$ and strong opponents reduce it. We examine which samples are strong proponents or opponents in VAE.

**Methodology.** We train separate VAE models on MNIST, CIFAR, and each CIFAR subclass. We then compute VAE-TracIn scores of all training samples over 128 test samples.

In MNIST experiments, we plot the distributions of influences according to whether training and test samples belong to the same class (See results on label zero in Figure 18 and full results in Figure 19). We then compare the influences of training over test samples to their distances in the latent space in Figure 20f. Quantitatively, we define samples that have the 0.1% highest/lowest influences as the strongest proponents/opponents. Then, we report the fraction of the strongest proponents/opponents that belong to the same class as the test sample and the statistics of pairwise distances in Table 2. Additional comparisons can be found in Appendix B.7,

In CIFAR and CIFAR subclass experiments, we compare influences of training over test samples to the norms of training samples in the latent space in Figure 22 and Figure 23. Quantitatively, we report the statistics of the norms in Table 3. Additional comparisons can be found in Appendix B.8.

**Results.** In MNIST experiments, we find many strong proponents and opponents of a test sample are its similar samples from the same class. In terms of class information, many ($\sim 80\%$) strongest proponents and many ($\sim 40\%$) strongest opponents have the same label as test samples. In terms of distances in the latent space, it is shown that the strongest proponents and opponents are close (thus

---

[3]AUC $\approx 1$ means the detection is near perfect, and AUC $\approx 0.5$ means the detection is near random.

Table 2: Statistics of influences, class information, and distances of train-test sample pairs in MNIST. "+" means top-0.1% strong proponents, "−" means top-0.1% strong opponents, and "all" means the train set. The first two rows are fractions of samples that belong to the same class as the test sample. The bottom three rows are means ± standard errors of latent space distances between train-test sample pairs.

| $d_{\text{latent}}$ | 64 | 96 | 128 |
|---|---|---|---|
| same class rate (+) | 81.9% | 84.0% | 82.1% |
| same class rate (−) | 37.3% | 43.3% | 40.3% |
| distances (+) | $0.94 \pm 0.53$ | $0.94 \pm 0.55$ | $0.76 \pm 0.51$ |
| distances (−) | $1.78 \pm 0.75$ | $1.84 \pm 0.78$ | $1.29 \pm 0.67$ |
| distances (all) | $2.54 \pm 0.90$ | $2.57 \pm 0.91$ | $2.23 \pm 0.92$ |

Table 3: The means ± standard errors of latent space norms of training samples in CIFAR and CIFAR-Airplane. Notations follow Table 2. It is shown that strong proponents tend to have very large norms.

| | | |
|---|---|---|
| CIFAR | (+) | $7.42 \pm 1.10$ |
| | (−) | $3.89 \pm 1.26$ |
| | (all) | $5.06 \pm 1.18$ |
| CIFAR-Airplane | (+) | $4.73 \pm 0.78$ |
| | (−) | $4.26 \pm 0.91$ |
| | (all) | $4.07 \pm 0.83$ |

Table 4: High level summary of influence functions in classical unsupervised learning methods ($k$-NN, KDE and GMM) and VAE. In terms of self influences, VAE is similar to KDE, a non-parametric method. In terms of proponents and opponents, VAE trained on MNIST is similar to GMM, a parametric method. In addition, VAE trained on CIFAR is similar to supervised methods [Hanawa et al., 2020, Barshan et al., 2020].

| Method | high self influence samples | low self influence samples |
|---|---|---|
| $k$-NN | in a cluster of size exactly $k$ | – |
| KDE | in low density (sparse) region | in high density region |
| GMM | far away to cluster centers | near cluster centers |
| VAE | large loss / visually complicated or high-contrast | small loss / simple shapes or simple background |

| Method | strong proponents | strong opponents |
|---|---|---|
| $k$-NN | $k$ nearest neighbours | other than $k$ nearest neighbours |
| KDE | nearest neighbours | farthest samples |
| GMM | nearest neighbours | possibly far away samples in the same class |
| VAE$_{(\text{MNIST})}$ | nearest neighbors in the same class | far away samples in the same class |
| VAE$_{(\text{CIFAR})}$ | large norms and similar colors | different colors |

similar) samples, while far away samples have small absolute influences. These findings are similar to GMM discussed in Section 3, where the strongest opponents may come from the same class (see Figure 7). The findings are also related to the supervised setting in the sense that dissimilar samples from a different class have small influences.

Results in CIFAR and CIFAR subclass experiments indicate strong proponents have large norms in the latent space. [4] This observation also happens to many instance-based interpretations in the supervised setting including classification methods [Hanawa et al., 2020] and logistic regression [Barshan et al., 2020], where large norm samples can impact a large region in the data space, so they are influential to many test samples.

**Visualization.** We visualize the strongest proponents and opponents in Figure 4. More visualizations can be found in Appendix B.7 and Appendix B.8. In the MNIST experiment, the strongest proponents look very similar to test samples. The strongest opponents are often the same but visually different digits. For example, the opponents of the test "two" have very different thickness and styles. In CIFAR and CIFAR subclass experiments, we find strong proponents seem to match the color of the images – including the background and the object – and they tend to have the same but brighter colors. Nevertheless, many proponents are from the same class. Strong opponents, on the other hand, tend to have very different colors as the test samples.

## 5.4 Discussion

VAE-TracIn provides rich information about instance-level interpretability in VAE. In terms of self influences, there is correlation between self influences and VAE losses. Visually, high self influence

---

[4]Large norm samples can be outliers, high-contrast samples, or very bright samples.

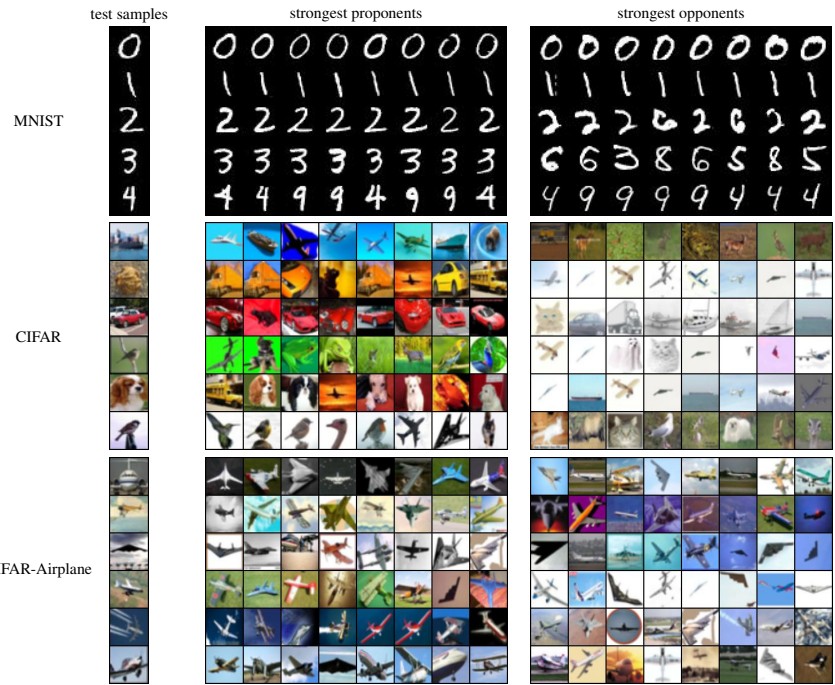

Figure 4: Test samples from several datasets, their strongest proponents, and strongest opponents. In MNIST the strongest proponents are visually similar while the strongest opponents are often the same digit but are visually different. In CIFAR and CIFAR-Airplane the strongest proponents seem to match the colors and are often very bright or high-contrast.

samples are ambiguous or high-contrast while low self influence samples are similar in shape or background. In terms of influences over test samples, for VAE trained on MNIST, many proponents and opponents are similar samples in the same class, and for VAE trained on CIFAR, proponents have large norms in the latent space. We summarize these high level intuitions of influence functions in VAE in Table 4. We observe there are strong connections between these findings and influence functions in KDE, GMM, classification and simple regression models.

## 6 Conclusion

Influence functions in unsupervised learning can reveal the most responsible training samples that increase the likelihood (or reduce the loss) of a particular test sample. In this paper, we investigate influence functions for several classical unsupervised learning methods and one deep generative model with extensive theoretical and empirical analysis. We present VAE-TracIn, a theoretical sound and computationally efficient algorithm that estimates influence functions for VAE, and evaluate it on real world datasets.

One limitation of our work is that it is still challenging to apply VAE-TracIn to modern, huge models trained on a large amount of data, which is an important future direction. There are several potential ways to scale up VAE-TracIn for large networks and datasets. First, we observe both positively and negatively influential samples (i.e. strong proponents and opponents) are similar to the test sample. Therefore, we could train an embedding space or a tree structure (such as the kd-tree) and then only compute VAE-TracIn values for similar samples. Second, because training at earlier epochs may be more effective than later epochs (as optimization is near convergence then), we could select a smaller but optimal subset of checkpoints to compute VAE-TracIn. Finally, we could use gradients of certain layers (e.g. the last fully-connected layer of the network as in Pruthi et al. [2020]).

Another important future direction is to investigate down-stream applications of VAE-TracIn such as detecting memorization or bias and performing data deletion or debugging.

## Acknowledgements

We thank NSF under IIS 1719133 and CNS 1804829 for research support. We thank Casey Meehan, Yao-Yuan Yang, and Mary Anne Smart for helpful feedback.

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
