# OpenReview forum: "Understanding Instance-based Interpretability of Variational Auto-Encoders"
_NeurIPS.cc/2021/Conference — NeurIPS 2021 Poster_

### Official Review · Reviewer_n5Rx · 2021-07-08

**Rating:** 6
**Confidence:** 4

**Summary:**

This paper provides the mathematical frame for influence functions (IFs) for several unsupervised machine learning approaches such as kNN, KDE, and GMM, and proposes a way to empirically compute VAE's IF, and performs several experiments to describe the rationale for using IF for unsupervised tasks. Their approach, VAE-TracIn, can answer which training samples are most responsible for increasing the likelihood of itself (i.e. self-influences) and/or a test sample. The authors claim that the self-influences can be used to discard undesired training samples before training and provide several experimental results showing that IFs for a test sample can identify most and least responsible training samples.


**Limitations And Societal Impact:**

The author briefly mentions in the conclusion that it is challenging to apply their approach to millions of samples.

**Main Review:**

- Originality: This paper proposes a way to calculate IFs for VAE and performs several experiments showing the ability of IFs and what they reveal about the impact of training samples. There is no big concern about originality.

- Quality and Significance: This paper provides experimental results describing how VAE-TracIn answers the three questions (L186-L189). However, an experiment should be performed in section 5.2 to evaluate how much can a VAE be improved by removing x% of samples with the highest self-influence from the training samples. This can explain why it is important to identify training samples with the highest and lowest self-influence training samples. Also, it is not clear what are the benefits of knowing which training samples have the highest/lowest influence over a test sample. What is the possible application of IFs over test data in unsupervised learning tasks?

- Clarity: The technical challenge and rationale are well described in the introduction. However, the benefit of using VAE-TracIn interpretation from the user's perspective is not well described in the introduction and abstract. The authors describe these in the later sections but I recommend the authors clearly describe the benefits and potential applications of VAE-TracIn in the introduction and/or abstract. Also, here are minor comments for improving the clarity.
  1. L49: Make italic or use different font styles for the terms "proponents" and "opponents" when they're first introduced.
  2. L84: Complimentary - Is this a typo?
  3. L363: Please update the journal/conference name.

- In summary, the main concern is that there is no or weak experimental evidence for illustrating the benefits and potential applications of using VAE-TracIn in training and test sample. The reviewer's score can be modified upon the authors' response and additional experimental results.

**Time Spent Reviewing:**

5

---

> ### Author Response · Authors · 2021-08-10
> **Response to Reviewer n5Rx**
>
> Thank you for the comments and for taking the time to review our work.
>
> - Quality and Significance
>
> Thank you for your suggestion on measuring how the VAE improves after removal of high self influence samples. We will add these experiments to the final version. As a preliminary experiment, we computed the average test loss of the VAE trained on datasets after removing $N_{rm}$ highest self-influence samples for MNIST. The losses are below; lower loss is better.
>
> |                           $N_{rm}$ |      0 |      4 |      8 |    32 |  128 |  512 |
> | ----------------------------------- | ------ | ------ | ------ | ------ | ------ | ------ |
> |Test loss ($\times10^{-3}$) | 4.24 | 4.21 | 4.18 | 4.18 | 4.17 | 4.18 |
>
> We see that the losses do decrease on average after removing the highest self-influence examples.
>
> We consider VAE-TracIn as a general-purpose tool that can potentially help us understand many things in the unsupervised setting. These can include detecting underlying memorization, bias or bugs (Feldman et al 2020) in unsupervised learning, performing data deletion (Asokan et al, 2020, Izzo et al 2021) in generative models, and examining training data with no label information.
>
> - Clarity
>
> It is a great suggestion to have a more detailed discussion of the benefits and potential applications in the introduction section. We will improve the writing accordingly. We will also address the minor comments.
>
> Please let us know if you have any more questions.

---

> > ### Comment · Reviewer_n5Rx · 2021-08-25
> > **Update the score**
> >
> > Thank you for describing the potential uses of VAE-TracIn and the benefits of knowing the influence of samples. I highly recommend adding them to the final version to emphasize the significance of this work. Also thank you for the additional experiments for evaluating how the VAE improves after removal of high self-influence samples. I updated the rating according to the authors' responses.

---

### Official Review · Reviewer_55DE · 2021-07-11

**Rating:** 7
**Confidence:** 5

**Summary:**

The paper characterizes influence estimation techniques to understand instance based interpretations for deep generative models.  Specifically, the authors carry out a comprehensive empirical and theoretical study about influence functions for generative models.


**Ethical Concerns:**

No ethical concerns

**Limitations And Societal Impact:**

Although the paper is a very good starting for instance based explanations for generative models, the paper has the following limitations:

(a) Experiments are relatively small scale on small networks/datasets. More experiments on modern VAEs will improve the paper from a practical standpoint.

(b) No solid discussion about how VAE-TracIn compares with other influence estimation technique (Koh et al. 2017). A small discussion would be useful to understand if the generally used influence technique used in supervised learning can be readily adapted for VAEs.


No negative societal impact.

**Main Review:**

Influence estimation techniques have not been studied well for unsupervised learning and this paper does a good job of filling that gap.  The authors primarily study the effectiveness of influence function based methods to understand example based explanations and data cleaning. In the paper, influence functions for classical unsupervised learning (e.g. KDE/GMM) is first studied where the authors provide closed form solutions for influence estimations. Then, to compute influence estimates for deep generative models,  instead of using the classical influence function (Koh et al. , Cook et al.), the authors use TracIn (Pruthi et al. ) and design it specifically for VAEs. With VAE-TracIn, the authors conduct multiple empirical studies to understand in details about self-influence and top/bottom influential samples.

Although the general influence estimation technique used in the paper (TracIn) is not entirely novel and borrowed from Pruthi et al. , the authors adapt it nicely for generative models which is not trivial to do. The empirical studies on VAE models give relevant information about what influential samples constitutes for in generative models (e.g. samples having high self-influence scores are generally atypical examples are important takeaways specifically for generative models and could be used in data cleaning). The related works section is comprehensive and the authors have covered most of the recent works. Overall, the paper is solid and can open up more research directions on using influence function based methods for generative models. However I have the following concerns/ questions to the authors:

(1) While the general influence function based method (Koh et al. 2017) is computationally expensive, it has a principled theory behind it. What is the relationship between VAE-TracIn and influence function? Would using both the methods give similar inferences?

(2) The experiments on VAE are on small networks and from a practical standpoint how can VAE-TracIn be scaled up for large networks ?

(3) How can VAE-TracIn be used for applications other than data cleaning?

(4) How precise is VAE-TracIn to quantify the ground-truth influence (e.g change in loss for a test sample when a training sample is removed)?


Although the experiments are relatively small scale, I feel this paper is a very good starting point for instance based interpretability for generative models and I lean towards an acceptance. I also hope the authors provide answers to (1)-(4) in the rebuttal.


**Time Spent Reviewing:**

3.5

---

> ### Author Response · Authors · 2021-08-10
> **Response to Reviewer 55DE**
>
> Thank you for the encouraging comments and for taking the time to review our work.
>
> First of all, we think your suggestion (4) is an excellent suggestion! Pruthi et al provided experiments that validate the approximation accuracy of TracIn. In their Appendix G, they look at the Pearson correlation coefficient between the TracIn scores and the change in loss for 100 test samples, and this coefficient is 0.978 (note that a coefficient = 1 means perfect correlation). Based on your suggestion, we repeated the same experiment for VAE-TracIn, and will add it to the final version. As a preliminary result, we have the Pearson correlation = 0.882 for CIFAR-Airplane (with 128 test samples).
>
> We then answer (1)-(3) below.
>
> (1) The influence function (Koh et al 2017) uses the second-order approximation to the change of loss under the assumption of convexity. TracIn (Pruthi et al 2020) uses the first-order approximation to the change of loss during the training process under the assumption that GD/SGD is the optimizer. We expect these methods to give similar results in the ideal situation. However, we implemented the method by Koh et al and found it to be inaccurate for VAE. A possible reason is that the Hessian vector product used to approximate the second order term is unstable. We will add some discussion on this to the final version.
>
> (2) The run-time complexity is linear in the number of samples, checkpoints, and network parameters. There are several potential ways to scale up VAE-TracIn for large networks and datasets. First, we observe both positively and negatively influential samples (i.e. strong proponents and opponents) are similar to the test sample. Therefore, we could train an embedding space or a tree structure (such as kd-tree) and then only compute VAE-TracIn values for similar samples. Second, because training at earlier epochs may be more effective than later epochs (as optimization is near convergence then), we could select a smaller but optimal subset of checkpoints to compute VAE-TracIn. Finally, we could use gradients of certain layers (e.g. the last fc layer as in Pruthi et al) of the network rather than all.
>
> (3) We consider VAE-TracIn as a general-purpose tool that can potentially help us understand many things in the unsupervised setting. These can include detecting underlying memorization, bias or bugs (Feldman et al 2020) in unsupervised learning, performing data deletion (Asokan et al, 2020, Izzo et al 2021) in generative models, and examining training data with no label information.
>
> “Limitations and Social Impact”: We will add a more detailed discussion of the relationship between related methods as discussed in (1) and scalability as discussed in (2).
>
> Please let us know if you have any more questions.

---

> > ### Comment · Reviewer_55DE · 2021-08-10
> > **Response to Authors**
> >
> > I thank the authors for their detailed response. While the technical novelty is slightly limited, I strongly believe this paper is a very good paper investigating alternate influence techniques to understand unsupervised learning. Considering there are very few works in this area, I strongly advocate for the acceptance of this paper.

---

### Official Review · Reviewer_yzrG · 2021-07-16

**Rating:** 6
**Confidence:** 3

**Summary:**

The paper applies the method TracIn to VAEs (beta-VAE) in order to estimate influence functions of training samples with new ones.

**Limitations And Societal Impact:**

No unstated limitations or societal concerns.

**Main Review:**

The novelty of the paper is not clear. Looking at the technical part, section 3 presents at length classical influence functions, but there is no novel component. Section 4 begins by explaining beta-VAEs for the most part, which is also background. Section 4.1 is where the new approach is explained, but is less than half a page long and, as far as I can tell, it is just straight application of TracIn.

The authors do outline as a unique challenge the impossibility to directly compute the expectation over the encoder Q, and proposed as a novel insight/method using samples to estimate the IF over this expectation (theorem 1). However, the estimation of the expectation over Q through sampling is just a standard feature of VAEs and the fact that this remains a valid approximation in estimating the IF instead of the ELBO directly derives from plugging in the ELBO into TracIn, and therefore it is not a technical novelty.

**Time Spent Reviewing:**

2

---

> ### Author Response · Authors · 2021-08-10
> **Response to Reviewer yzrG**
>
> Thank you for the comments and for taking the time to review our work.
>
> Our novel contributions in this paper are (a) we propose a framework for and develop an instance-based interpretation method for unsupervised learning based on influence functions and TracIn; the adaptation to the unsupervised setting is non-trivial (as also observed by Reviewer FRdn and 55DE).
>
> (b) We provide novel theoretical analysis of influence functions for other unsupervised learning methods (Gaussian mixture models and KDEs) in section 3 and A.1. We then provide sample complexity theory of influence functions for VAEs in section 4 (for conciseness, we put the complete theory in section A.2). These were not done before.
>
> (c) We evaluate VAE-TracIn on real data. We analyze self and test influences, obtain several interesting empirical observations, and show potential application of the proposed method as a general-purpose solution to understanding unsupervised models.
>
> Please let us know if you have any more questions.

---

> > ### Comment · Reviewer_yzrG · 2021-08-11
> > **Rebuttal**
> >
> > I see that the supplementary contains valuable analysis on the use of influence functions for unsupervised models. This is certainly of value and novel and therefore I will raise my score.
> >
> > I still feel that these analyses and their conclusion could be rendered better in the main body:
> >
> > I see now that section 3 and 4 look at the same topic; influence functions in unsupervised models (first in simple models and then in deeper VAEs) and analyse theoretically their behaviour. However, they do feel very disconnected and the fact that all experiments implement only the method of section 4 make section 3 seem unrelated and almost background. Summarising some general points could help:
> > 1) Given the analyses of supplementary A, Is there any general new insight about influence functions in unsupervised learning that derived from the analysis of both simple models and trac-VAE?
> > 2)Table 1 nicely summarised what different influence functions quantify influence; is trac-VAE similar to any of them, on a high level? does it avoid some undesired effect/mis-quantification the other methods suffer from?

---

> > > ### Author Response · Authors · 2021-08-12
> > > **Response to Reviewer yzrG**
> > >
> > > Thank you very much for your follow-up comments. We would like to mention that we have a small experiment that compares the different classical methods in Appendix B.1; but we agree that it needs to be better highlighted in the main body. We will do this in an updated version of the paper.
> > >
> > >
> > > Regarding your question 1, here are the insights we obtain on influence functions in unsupervised learning vs. supervised learning:
> > >
> > > - Self influences: In supervised learning, Pruthi et al. find high self influence samples come from densely populated areas while low self influence samples come from sparsely populated areas. We find that in unsupervised learning, the opposite holds -- high self influence samples are outliers and low self influence samples are from high density areas.
> > >
> > > - Strong proponents: It has been observed by several papers that in supervised learning, strong proponents have large norms and are outliers. In unsupervised learning, we find this to be **not** true for most methods (e.g. classical methods and VAEs trained on MNIST, where strong proponents are nearby samples) except for VAEs trained on CIFAR.
> > >
> > >
> > > Regarding your question 2, we think this is an excellent suggestion -- thank you!  We have discussed some high-level observations for VAEs in section 5.4, but to make the results more visible, we will re-organize and append a row about VAEs to Table 1. Below is the summary of these observations for VAEs:
> > >
> > > - Self influences: high self influence samples have large loss and are visually complicated or high-contrast. Low self influence samples have small loss and share similar visual shapes or background. These are similar to KDE in Table 1.
> > >
> > > - Proponents & opponents: for VAEs trained on MNIST, many proponents and opponents are samples from the same class, and proponents are closer to test samples. This is similar to GMM in Table 1. For VAEs trained on CIFAR, strong proponents have large norms; they match the colors of test samples and are brighter. This is similar to supervised learning methods.
> > >
> > > Please let us know if you have any more questions!

---

### Official Review · Reviewer_FRdn · 2021-07-16

**Rating:** 6
**Confidence:** 3

**Summary:**

The paper investigates influence functions for several classical unsupervised learning methods and VAE. Influence functions in unsupervised learning can reveal the most responsible training samples that increase the likelihood (or reduce the loss) of a particular test sample. A particular usage of the proposed approach is for data cleaning. The authors demonstrated their approach on the MNIST and CIFAR-10 datasets.

**Limitations And Societal Impact:**

See my comments in "Main Review".

**Main Review:**

I am not an expert on instance-based interpretation methods so I cannot evaluate the significance of this work. But the methods proposed here look useful to me. The extension of instance-based interpretation from supervised learning models to unsupervised learning models is non-trivial. A potential drawback of this work is that the authors do not validate their experimental results with other existing data cleaning methods. Therefore, I am not sure the approach proposed here provides meaningful practical usage in data cleaning. I, therefore, refer to other reviewers on the evaluation of the significance of this work.

**Time Spent Reviewing:**

1

---

> ### Author Response · Authors · 2021-08-10
> **Response to Reviewer FRdn**
>
> Thank you for the comments and for taking the time to review our work.
>
> The main contribution of our paper is to build instance-based interpretation for unsupervised models, which is non-trivial as you mentioned. We consider such interpretation as a general-purpose tool that can potentially help us understand many things in the unsupervised setting. These can include detecting underlying memorization, bias or bugs (Feldman et al 2020) in unsupervised learning, performing data deletion (Asokan et al, 2020, Izzo et al 2021) in generative models, and examining training data with no label information.
>
> As for the data cleaning application, it is not our main goal to beat the best data cleaning method in the literature. It is likely that the best cleaning method is not based on influence functions; instead, we believe that methods specifically optimized for data cleaning (such as anomaly detection) can have better results. However, we provide this example to show the potential of our interpretation as a general-purpose solution in understanding unsupervised problems as discussed above.
>
> Please let us know if you have any more questions.

---

### Decision · Program_Chairs · 2021-09-27

**Decision:**

Accept (Poster)

**Comment:**

The paper investigates the instance-based interpretation method based on influence functions in the Variational Auto-Encoder framework. The paper is well-organized and rather easy to follow. The reviewers tend to agree on the positive aspects of the paper, such as:
- Even though some ideas were introduced elsewhere, the application to VAEs is novel and interesting.
- The paper is well-positioned in the literature review.

The main disadvantage of the paper is the experimental part:
- The paper does not compare their experimental results with other existing data cleaning methods.
- There is little experimental evidence for illustrating the benefits and potential applications of using VAE-TracIn in training and test samples.

Overall, all reviews have a tendency towards the acceptance with one reviewer being positive about the acceptance. Therefore, I believe that the paper could be accepted.